# New Conservative Approach for the Management of Recurrent Sublingual Ranula—A Case Report

**DOI:** 10.3390/ijerph20032398

**Published:** 2023-01-29

**Authors:** Antonino Fiorino, Edoardo Staderini, Rosalba Diana, Carlo Rengo, Patrizia Gallenzi

**Affiliations:** 1Department of Neuroscience and Reproductive and Odontostomatological Sciences, “Federico II” University of Naples, 80131 Naples, Italy; 2Postgraduate School of Orthodontics, Università Cattolica del Sacro Cuore, Largo Agostino Gemelli 8, 00168 Rome, Italy; 3Department of Medicine, Surgery and Dentistry, University of Salerno, 84081 Salerno, Italy; 4UOC Odontoiatria Generale e Ortodonzia, Dipartimento di Testa-Collo e Organi di Senso, Fondazione Policlinico Universitario “A. Gemelli” IRCCS, Largo Agostino Gemelli 8, 00168 Rome, Italy

**Keywords:** ranula, mucocele, marsupialization, Hybenx^®^

## Abstract

Ranula is a rare and benign extravasation mucocele that is clinically characterized by asymptomatic sublingual or submandibular masses. Surgical excision is considered the most effective treatment approach, but it has been associated with high invasiveness and several complications (hemorrhage, damage to Wharton’s duct, and lingual nerve injury). Over the past decade, more conservative therapies have been rapidly disseminated into clinical practice to seek a more effective and less traumatic approach for young patients. In this report, an 8-year-old female with an asymptomatic, recurrent sublingual ranula was treated using a conservative approach with marsupialization and an intracystic injection of a plaque remover (Hybenx^®^ gel). After incision of the cystic dome, Hybenx^®^ gel was applied into the cystic lumen for 20 seconds and then aspirated; next, the area was rinsed thoroughly with sterile saline solution before suturing. Ultrasound re-evaluation at 10 months and intraoral clinical examination at 24 months confirmed the absence of relapse. Our results support the hypothesis that marsupialization combined with intracystic injection of Hybenx^®^ gel could be an encouraging conservative treatment alternative for recurrent sublingual ranula in children. Further randomized controlled trials are needed to test this hypothesis.

## 1. Introduction

Ranula is a rare, benign, congenital or acquired extravasation mucocele (pseudocyst) that is usually located in the region above the mylohyoid muscle (sublingual ranula) or in the submandibular space (*plunging ranula*) [1,2]. The prevalence of sublingual ranula is 0.02%, whereas plunging ranula has an overall annual crude incidence of 2.4 per 100,000 person-years [3,4].

Sublingual ranula frequently occurs in the first and second decade of life and is more common in females (M/F ratio 1:1.4); *plunging ranula* has a peak incidence during the third decade of life, being rare in childhood, and is more prevalent in females [3,4,5].

The etiology of ranula is related either to congenital malformation (anatomic anomalies of the mylohyoid muscle), which causes improper drainage of the sublingual glands, or traumatic and obstructive events involving the glands or their excretory ducts [6,7,8]. The pathogenesis is related to the extravasation of saliva from the sublingual gland into the surrounding tissues, such as periglandular and periductal tissues [9,10].

Ranula appears as a blue, dome-shaped fluctuant mass on the floor of the mouth; however, deeper lesions may hide beneath the overlying intact mucosa. As an extravasation type mucocele, ranulas lack a true cyst wall, as they are lined with granulation tissue rather than epithelium [11,12].

Due to its location, the differential diagnosis includes a true salivary duct cyst, dermoid cyst, and cystic hygroma. The diagnostic process involves the clinical observation of phenotypic features (site, size—volume, shape—regular/irregular, surface—irregular/smooth, local swelling, hemorrhagic content), palpation of the mass (tenderness, consistency—soft/hard, mobility), radiographic imaging (ultrasonography, magnetic resonance imaging, or computerized tomography), and histopathological analysis of the cystic content (needle aspiration biopsy) [13,14].

The choice of appropriate treatment is established based on the size, location, and origin of the lesion [15]. Based on the current literature, the most adopted treatment modalities can be divided into two main categories: surgical and conservative approaches. The surgical approach is an excisional biopsy (with or without using a laser) involving an incision and complete removal of the cystic lesion. A recent meta-analysis by Chung et al., determined that surgical removal of the gland was the best curative treatment option (98.84% success rate) [16].

However, surgical management is associated with invasiveness and postoperative morbidity due to the position of the critical anatomical structures in the sublingual and submandibular areas; indeed, it could be associated with lingual nerve injury, damage to Wharton’s duct, and hemorrhagic episodes [16,17]. Due to suboptimal cooperation among young patients, surgical treatment is often performed under general anesthesia, thus providing an additional procedural risk. Over the past decade, more conservative therapies have rapidly disseminated into clinical practice to seek more effective and less traumatic approaches for young patients [16].

These conservative approaches involve incision of the cystic dome, aspiration of the cystic fluid, curettage of the inflammatory tissue (if present), and progressive drainage of the cystic lesion through sequential medications. These treatment approaches apply many techniques, including sclerotherapy (injection of OK-432; Chugai Pharmaceutical Co., Ltd., Chuo, Tokyo, Japan), marsupialization, and micromarsupialization [2,5]. The success rates of these conservative procedures were 76.44%, 78.72%, and 93.85%, respectively [16].

Sclerotherapy consists of an intralesional injection of a sclerosing agent, such as Picibanil. This procedure induces a local inflammatory reaction, leading to a reduction in fluid production, collapse of the walls, and, finally, resolution of the lesion. Due to its therapeutic potential, the use of Picibanil was first introduced for the treatment of lymphatic malformations [18,19].

Although it is not possible to obtain a resolution in 100% of cases [20], sclerotherapy with OK-432 may be indicated as a primary conservative treatment for sublingual ranulas, but there is no consensus on the management of plunging ranulas [19,21].

Marsupialization is a conservative technique based on the surgical removal of the dome of the ranula and connecting the epithelial wall of the pseudocystic lesion to the surrounding healthy oral mucosa. Removal of the roof of the lesion causes cystic decompression and shrinkage through leakage of cystic fluid. Occasionally, a drainage device is inserted into the lesion after bubble dome removal and sutured to avoid closure of the ostium. Then, the deeper portion of the cyst can heal by secondary intention [15,22]. Concerning clinical indications, this procedure is usually applied to large mucoceles in adults to avoid damage to surrounding anatomic structures [23].

Micromarsupialization is a minimally invasive approach carried out to drain the accumulated mucin and to allow the formation of an epithelial tract between the intraoral epithelial surface and the underlying glandular tissue. The procedure is performed by passing a silk suture on the dome of the cyst along the largest diameter of the lesion. Concerning clinical indications, micromarsupialization is indicated for sublingual ranulas, as well as for smaller mucoceles located on the lower lip or the ventral surface of the tongue [24].

The abovementioned conservative approaches (sclerotherapy, marsupialization, and micromarsupialization) can be considered minimally invasive treatment options due to shorter procedural time, minimal discomfort, and absence of postoperative complications [2,5].

Hybenx^®^ Oral Decontaminant (EPIEN Medical Inc., St. Paul, MN, USA) is used in various settings, such as symptomatic treatment of recurrent aphthous stomatitis or as a bacterial plaque remover in peri-implantitis treatment, as it can decontaminate the exposed vital pulp and endodontic system [25,26]. According to the manufacturer’s instructions for use, the action of Hybenx^®^ is based on dehydrating organic tissues and biofilms due to the fast-acting principle of “Desiccation Shock Technology^®^” of sulphonated phenols; sulphonated phenols remove water from the combination of infectious microbial material, dead cellular debris (biofilm), and macromolecules found in an infectious lesion. This effect denatures (destroys) the molecular structure of this infectious material and loosens its attachment to the lesion surface. This process occurs within less than a minute after contact [25,26].

It can be assumed that the rapid action of Hybenx^®^ removes superficial plaque biofilm, debrides loose necrotic debris, and lessens the risk of biofilm contamination of the wound area. However, there is a lack of literature regarding the use of dehydrating agents (such as Hybenx^®^) as alternatives to Picibanil (OK-432). To the best of our knowledge, the use of Hybenx^®^ gel for the treatment of pseudocystic lesions is a new approach that does not refer to existing publications using the same method for the same purpose. The scientific rationale that led the authors to treat a relapsing mucocele with Hybenx^®^ gel derives from the possibility of resorting to a similar but more conservative and less invasive procedure than sclerotherapy [5]. Its action is based on dehydrating organic tissues and biofilms due to the action of sulphonated phenols [27].

It can be assumed that the rapid action of Hybenx^®^ may destroy inflammatory mucoceles by directly attacking the glandular mass responsible for extravasation. However, there is a lack of literature regarding the use of dehydrating agents (such as Hybenx^®^) as alternatives to OK-432.

Therefore, this case report discusses 2-year follow-up outcomes of the marsupialization and topical application of a biofilm remover (Hybenx^®^) for a sublingual ranula in a pediatric patient.

## 2. Materials and Methods

The study protocol was approved by the Institutional Review Board (n. 0037733/22, Appendix A). An 8-year-old female patient was referred to the Fondazione Policlinico A. Gemelli teaching hospital for evaluation of a sublingual swelling. Examination revealed the presence of a mass in the right sublingual area that was asymptomatic upon palpation. The surface was covered by a translucent dome with small hemorrhagic areas (Figure 1a). Removal of the dome resulted in the release of a transparent saliva-like liquid (Figure 1b).

The parents reported that the cystic volume changed cyclically with intervals of 6–10 days. The patient did not report any pain but only slight superficial discomfort in the days preceding the dome rupture due to tension on the surrounding tissues.

Clinical evaluation of salivary flow through the excretory ducts of the submandibular and sublingual glands appeared normal. The patient’s past medical history was unremarkable. Dental examination showed a first molar class, dental crowding with malposition of tooth 3.1, and moderate rotation of the upper central incisors. From the orthopantomographic radiographic examination, no sialolithiasis was evident.

A submandibular and intraoral sublingual ultrasound examination was performed at the Head and Neck Radio-diagnostic Center of the Department of Radiology at the Integrated University Hospital in Verona, Italy.

An enlargement of the right sublingual gland body (11 mm × 8 mm) with a well-defined wall was observed. No other alterations affecting the ipsilateral or contralateral salivary glands were appreciable.

Based on the patient’s anamnesis and clinical signs, a suspected diagnosis of sublingual (simple) ranula originating from the sublingual gland was hypothesized.

Based on the ultrasound examination,, the radiologist declared that the radiographic findings confirmed the diagnostic suspicion of sublingual ranula (Figure 2a,b).

Considering the suspected nature of the lesion, its extension, and the patient’s cooperativeness, it was decided to proceed with marsupialization. After two unsuccessful marsupialization attempts using a catheter (a small rubber tube derived from a Lip Bumper/Labial Arch. 0.7/0.8 mm k0470-00— Leone SpA, Sesto Fiorentino, Florence, Italy) (Figure 3 and Figure 4) and gauze packing of the open surgical wound (Figure 5a–c) over a period of 3 months, the clinician (A.F.) decided to use a chemical approach to reduce intra- and postoperative patient discomfort. This study was conducted in compliance with the Ethical Principles for Medical Research Involving Human Subjects outlined in the Declaration of Helsinki in 1975; written informed consent for the procedure and use of personal data was obtained from both parents before the intervention.

The procedure was as follows:Anesthesia. Premedication with topical anesthesia (Xylonor Spray—10% lidocaine analgesic solution; Septodont) was followed by infiltration with mepivacaine (Pierrel S.p.A. mepivacaine 20 mg/mL and adrenaline 1:100,000) around the margins of the lesion.Incision. Soft tissue incisions with microsurgical scissors to remove the pseudocystic dome and provide adequate access to the area of interest.Probing and irrigation. The lumen of the lesion was probed, irrigated with sterile saline solution, and dried with cotton gauze.Application of Hybenx^®^. Hybenx^®^ gel was injected into the lumen of the lesion. The choice of gel was based on its consistency to avoid dispersion of the product into the surrounding tissues. The gel (0.5 mL) was allowed to act for approximately 20 s and then aspirated, and the area was rinsed with copious amounts of saline solution. After complete and careful gel removal, the lesion margins appeared opaque white, with no signs of bleeding.Suturing. To delay closure of the ostium and inversion of the margins of the lesion, horizontal mattress sutures (6-0 coated Vicryl^®^ undyed 1 × 18″ s-14 double armed—Ethicon^®^) were performed to evert the wound edges and keep the ostium open (Figure 5).

The patient was monitored for several minutes after the application of Hybenx^®^ gel and contacted a few hours later to ascertain her health condition. The patient was asked not to consume solid food for 2 days, keep the area at rest as much as possible, and use 0.2% chlorhexidine mouthwash for 1 minute twice a day for 6 days. No antibiotic therapy was prescribed. In the case of pain, the patient could take paracetamol (500 mg tablets) up to three times a day.

## 3. Results

The patient reported mild discomfort only on the first day and took paracetamol twice a day. No pain was reported in the following days. Upon clinical examination, the lesion showed good signs of healing with clear reduction in extension and depth.

At the 2-week follow-up, the formation had not relapsed; healing of the lesion and mucosal integrity were also clinically observed, without any negative postoperative symptoms or complications. No clinical signs of recurrence were noticed upon inspection or palpation during her follow-up visits (Figure 6).

After 10 months, the ultrasound examination was repeated and showed complete shrinkage of the lesion without recurrence; the volume of the right sublingual gland was equal to that of the contralateral sublingual gland (Figure 7).

No further symptoms were reported, and no postoperative complications were identified. At the two-year follow-up, clinical examination confirmed the “complete’’ absence of a sublingual ranula (Figure 8), defined as absence of a lesion identifiable on inspection, palpation, or imaging [28].

## 4. Discussion

This report presents a case of a recurrent sublingual ranula treated with a conservative approach based on a single intracystic application of Hybenx^®^ gel. The procedure was demonstrated to be easy and minimally invasive. The patient experienced less discomfort with this therapeutic modality than after previous treatments (Phases 1 and 2). No negative postoperative symptoms or complications were observed.

The use of Hybenx^®^ gel is intended to mimic the action of OK-432 (Picibanil) without having the same adverse effects. OK-432 was one of the first medications proposed for the conservative treatment of cystic lesions [29,30]. It is a lyophilized mixture of a low virulence strain of *Streptococcus pyogenes* that has been incubated with benzylpenicillin.

As clarified in the introduction, the use of Picibanil has been widely documented for the treatment of head and neck lymphangiomas due to its ability to produce selective fibrosis. Despite encouraging data about success rates (76.44–78.2%) [16] reported in the literature, intracystic injection of OK-432 has been associated with adverse effects, including fever, local inflammation, transient increase in blood platelet concentration, and sclerosis of glandular epithelial tissue [29,30]. However, these complications could be considered less significant than those associated with the conventional surgical approach [31,32].

Our case report documents the effectiveness, safety, and feasibility of using Hybenx^®^ gel to remove a recurrent sublingual ranula in a pediatric patient. The advantage of Hybenx^®^ gel is the availability of the equipment in the dental care office, as it has a variety of different purposes. Moreover, the clinical application is easy (the gel flows directly into the pseudocystic space), short (Hybenx^®^ gel is applied for approximately 20 s), and minimally invasive (the application of Hybenx^®^ gel requires only a small incision, so there is minimal postoperative discomfort). Compared to marsupialization, the adjunct application of Hybenx^®^ gel provides additional denaturing action against biofilms and tissue debris without impacting other surrounding areas. Compared to micromarsupialization, which does not require any surgical incision, our procedure provides the opportunity to remove and obtain the dome of the pseudocyst for histological analysis. Moreover, micromarsupialization carries the risk of early spontaneous loss of the suture thread, requiring a new intervention [24]. Compared to sclerotherapy, the reported side effects of OK-432 include fever, odynophagia, local inflammation, and swelling [33]. The patient in this case did not show these side effects and only had slight local discomfort.

Even if the surgical approach is associated with lower recurrence rates, salivary gland exeresis has been associated with lingual nerve injury, damage to Wharton’s duct leading to stenosis, obstructive sialadenitis, leakage of saliva, and even hemorrhagic episodes in the pediatric population [16,22,34]. In young and poorly cooperative patients, surgery should be performed under general anesthesia, which has additional perioperative risks and a greater burden of care.

To the best of our knowledge, this is the first study analyzing Hybenx^®^ gel in the treatment of a sublingual ranula.

The main limitation of this study is that it is a case report. The success rate and related predictive factors for the efficacy of this technique should be further investigated with randomized controlled clinical trials, enrolling patients from different age groups.

## 5. Conclusions

This case report shows the effectiveness, safety, and ease of application of Hybenx^®^ gel in the conservative management of a sublingual ranula in a child. The advantages of this technique are shorter surgical time and low postoperative morbidity.

## Figures and Tables

**Figure 1 ijerph-20-02398-f001:**
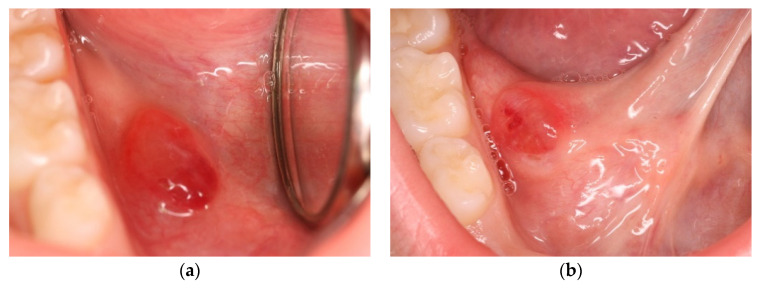
Intraoral physical examination. (**a**) Presence of pseudocystic lesion on the right side of the floor of the mouth. (**b**) Clinical evaluation during the tongue elevation phase.

**Figure 2 ijerph-20-02398-f002:**
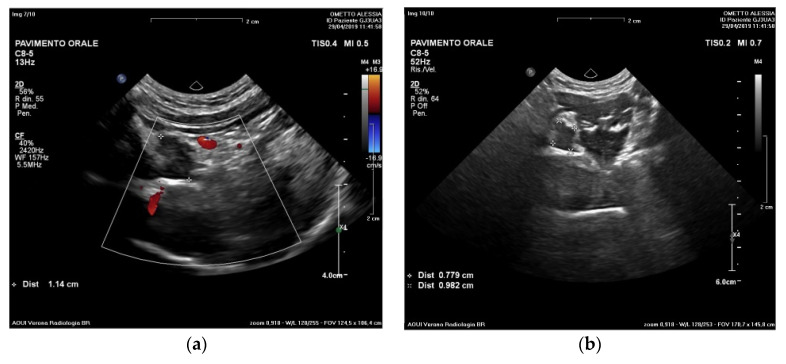
Ultrasound examination. Evaluation at different frequencies (**a**) and size (**b**) of the pseudocystic lesion. A hypodense lesion (11 × 8 mm) was observed.

**Figure 3 ijerph-20-02398-f003:**
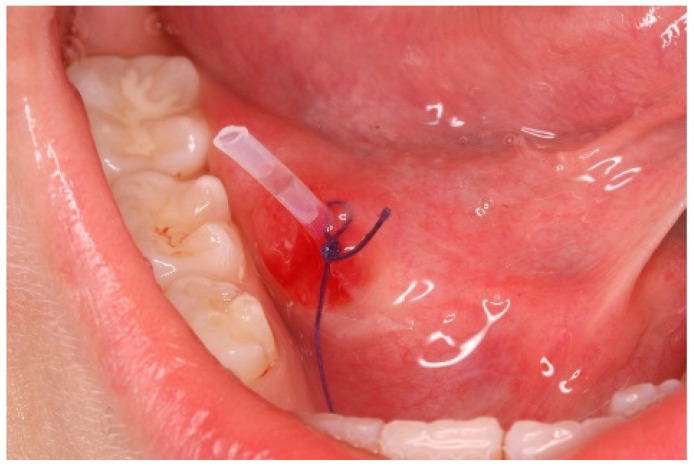
This figure shows the marsupialization technique (attempt n° 1): rubber drainage tube inserted into the lesion after bubble dome removal and sutured to avoid ostium closure.

**Figure 4 ijerph-20-02398-f004:**
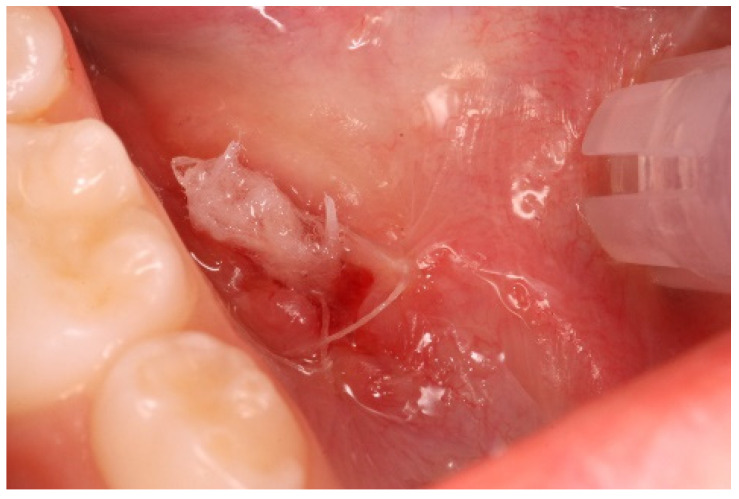
This figure shows the marsupialization technique (attempt n° 2): gauze is inserted into the lesion after bubble dome removal and sutured.

**Figure 5 ijerph-20-02398-f005:**
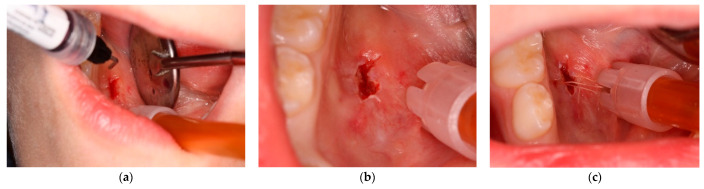
This figure shows modified sclerotherapy using Hybenx^®^ gel (attempt n° 3). (**a**) Hybenx^®^ gel is injected into the lumen of the lesion; (**b**) 20 s after application, Hybenx^®^ gel is aspirated, and the area is rinsed with saline solution; (**c**) the margins of the lesion are sutured.

**Figure 6 ijerph-20-02398-f006:**
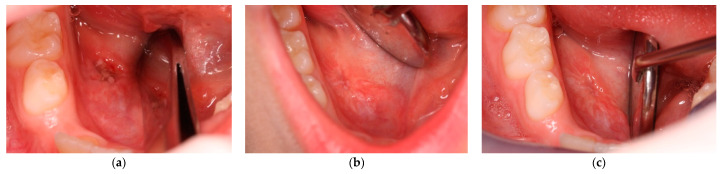
Clinical re-evaluation at (**a**) 48 h; (**b**) 7 days; and (**c**) 12 months of follow-up.

**Figure 7 ijerph-20-02398-f007:**
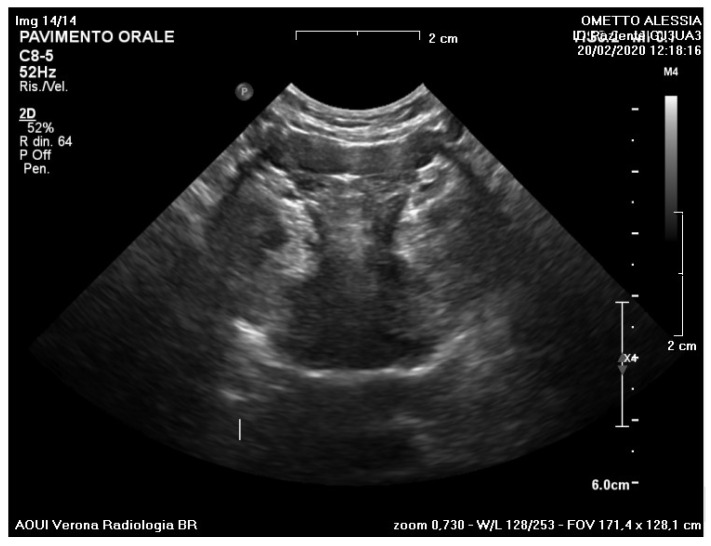
Ultrasound re-evaluation at 10 months showed complete resolution of the lesion.

**Figure 8 ijerph-20-02398-f008:**
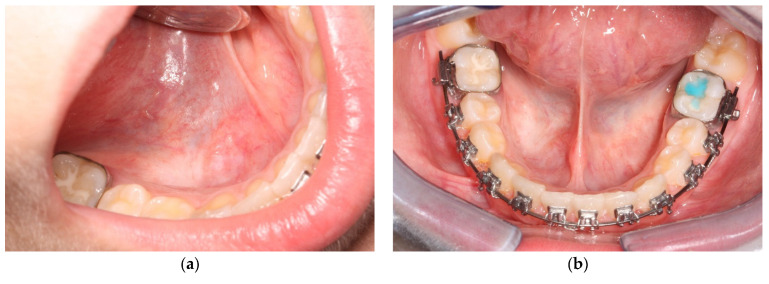
Intraoral clinical examination at 24-month follow-up. Right side (**a**) and front view (**b**).

## Data Availability

All data generated or analyzed during this study are available from the corresponding author on reasonable request.

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
