# Peer review of "New Conservative Approach for the Management of Recurrent Sublingual Ranula—A Case Report"

_ijerph, 2023, doi:10.3390/ijerph20032398_

Round 1

Reviewer 1 Report (Previous Reviewer 3)

Dear Authors, the correction improved the quality of the article.

Author Response

We thank the reviewer for appreciating our effort for improving the paper. 

Reviewer 2 Report (New Reviewer)

I have read this paper carefully and have the following observations to make:

·      Abstract: The conclusions that the authors draw from just one clinical case are too ambitious. It is not possible to assess whether a certain material is safe and effective with just one clinical case.

·      Introduction: what is the prevalence of plunging ranula? The authors must describe the indications for carrying out each type of treatment protocol (sclerotherapy, marsupialization, micromarsupialization).

·      Material and Methods: The ethical committee should appear at the top of the material and methods section. please explain points (a) and b)) of Figures 1, 2, and 8.

·      Results: in the abstract, the authors refer to a follow-up of 2 years, but in the results section, they only present up to 10 months. the authors must present the missing results.

·      Discussion: authors should discuss the advantages of this conservative method compared to other available conservative methods. There are other ways to overcome the limitations of this study, for example, randomization of the sample. Authors should develop this topic.

·      Conclusion: presents too much information for a conclusion. I suggest keeping only the 1st and 4th sentences.

Author Response

Comments to the Authors:

-"  have read this paper carefully and have the following observations to make:

  • Abstract: The conclusions that the authors draw from just one clinical case are too ambitious. It is not possible to assess whether a certain material is safe and effective with just one clinical case.”

Response to the Reviewer:

We thank the reviewer for the suggestion. As suggested, we modified the conclusion of the abstract.

Comments to the Authors:

- “Introduction: what is the prevalence of plunging ranula? The authors must describe the indications for carrying out each type of treatment protocol (sclerotherapy, marsupialization, micromarsupialization).”

Response to the Reviewer:

We thank the reviewer to take this aspect up. We added the prevalence of the plunging ranula and the indications of the conservative treatment strategies.

Comments to the Authors:

- “Material and Methods: The ethical committee should appear at the top of the material and methods section. please explain points (a) and b)) of Figures 1, 2, and 8.”

Response to the Reviewer:

We thank the reviewer to take this aspect up. We moved the ethical committee approval at the top of the section and added the comments on Figures 1,2, and 8 in the revised manuscript.

Comments to the Authors:

- “Results: in the abstract, the authors refer to a follow-up of 2 years, but in the results section, they only present up to 10 months. the authors must present the missing results.”

Response to the Reviewer:

We thank the reviewer to take this aspect up. We performed ultrasound re-evaluation at 10 months and the intraoral clinical examination at 24 months. The follow-up protocol has been clarified in the abstract section.

Comments to the Authors:

- “Discussion: authors should discuss the advantages of this conservative method compared to other available conservative methods. There are other ways to overcome the limitations of this study, for example, randomization of the sample. Authors should develop this topic.”

Response to the Reviewer:

Point taken. We have discussed the advantages of the novel approach in comparison with other conservative alternatives. We have also added the randomization issue in the limitations of the study.

Comments to the Authors:

- “Conclusion: presents too much information for a conclusion. I suggest keeping only the 1st and 4th sentences.”

Response to the Reviewer:

We thank the reviewer for the kind suggestion. The conclusion has been modified in the revised manuscript.

Round 2

Reviewer 2 Report (New Reviewer)

Accept in present form

This manuscript is a resubmission of an earlier submission. The following is a list of the peer review reports and author responses from that submission.

Round 1

Reviewer 1 Report

Dear authors,

I reviewed the article titled “New conservative approach for the management of recurrent sublingual ranula - a case report”.

I considered it interesting however with a low level of novelty and scientific significance.

The abstract did not present the methodology used and it is incomplete.

In the intro, before the objective of the study, there is a “OK - 432”.

The study used ultrasound which is interesting. Therefore, where is in the literature the application of Hybenx gel intracystic?

How the authors can consider the application safe?

Can the authors include a scientific-based evidence to support this type of treatment or the use was empirical?

Author Response

Comments to the Authors:

-"I reviewed the article titled “New conservative approach for the management of recurrent sublingual ranula - a case report”.

I considered it interesting however with a low level of novelty and scientific significance.

The abstract did not present the methodology used and it is incomplete..”

Response to the Reviewer:

We thank the reviewer to find our paper interesting. The methodology has been added in the revised abstract.

Comments to the Authors:

-"In the intro, before the objective of the study, there is a “OK - 432”.

Response to the Reviewer:

Point taken. We have specified the abbreviation in the revised manuscript.

Comments to the Authors:

-" The study used ultrasound which is interesting. Therefore, where is in the literature the application of Hybenx gel intracystic?”.

Response to the Reviewer:

The novelty of this approach consists in the absence of publications regarding the use of Hybenx for the treatment of pseudocystic lesions.

Existing publications refer to the following uses:

  1. a) Decontamination of supragingival tooth surfaces (direct application on the tooth / root surface) (A Topical Desiccant Agent in Association with Ultrasonic Debridement in the Initial Treatment of Chronic Periodontitis: a Clinical and Microbiological Study. New Microbiologica 38: 393-407. 2015. G Lombardo et all)
  2. b) Decontamination of subgingival tooth surfaces (infiltration into the periodontal pocket) (The effects of a desiccant agent in the treatment of chronic periodontitis: a randomized, controlled clinical trial. J Clin Oral Invest. 17 June 2017 (doi:10.1007/s00784-017-2154-7). G Isola et all)
  3. c) Treatment of perimucositis (direct infiltration into the peri implant pockets) (The Treatment of Peri-Implant Diseases: A New Approach Using HYBENX® as a Decontaminant for Implant Surface and Oral Tissues. ORAL & Implantology 9(3):106-114. 2016. MA Lopez et all)
  4. d) Decontamination of root canals in endodontic therapy (direct infiltration into endodontic system) (THE EFFECTS OF A NEW DECONTAMINANT SOLUTION ON ROOT CANAL BLEEDING DURING ENDODONTIC TREATMENT: A RANDOMIZED CONTROLLED STUDY. Pace L. et all)
  5. e) Treatment of recurrent aphthous lesions (direct application on the mucosal lesions) (Randomized controlled trial of the efficacy of HYBENX in the symptomatic treatment of recurrent aphthous stomatitis. J. Oral Diseases 15, 155–161. 2009. SR Porter, K Al-Johani, S Fedele, and DR Moles)

Comments to the Authors:

-"How the authors can consider the application safe?”.

Response to the Reviewer:

The safety of Hybenx® gel, respecting the times and methods of application, is certified by the manufacturer. In this case, the same method recommended by the manufacturer for the decontamination of the subgingival root surfaces was used. Then, the product was introduced into the lumen of the pseudocyst, previously irrigated and probed, and eliminated with abundant rinsing after the times recommended by the manufacturer. The patient was monitored for several minutes after the application of Hybenx® and contacted a few hours later to ascertain her health conditions.

At no time any kind of adverse events were observed or reported.

Furthermore, the author F.A. is experienced with the use of Hybenx® gel, as reported from previous studies:

- Full-mouth ultrasonic debridement in combination with HYBENX® Oral Tissue Decontaminant as an initial approach to the treatment of chronic periodontitis. (Translation from the original Italian Manuscript). Quintessence International 28(3): 5-14. 2012. G Lombardo, A Pardo, A Fiorino, G Corrocher, and G Urbani.

Comments to the Authors:

-"Can the authors include a scientific-based evidence to support this type of treatment or the use was empirical?”.

Response to the Reviewer:

The use of Hybenx® gel for the treatment of pseudocystic lesions is a novelty that does not refer to existing publications that use the same method for the same purpose. The scientific rationale that led the authors to treat a relapsing mucocele with Hybenx® gel derives from the possibility of resorting to a similar but more conservative and less invasive procedure than sclerotherapy (Bowers, E.M.R.; Schaitkin, B. Management of Mucoceles, Sialoceles, and Ranulas. Otolaryngologic clinics of North America 2021, 54, 543-551, doi:10.1016/j.otc.2021.03.002.

Bachesk, A.B.; Bin, L.R.; Iwaki, I.V.; Iwaki Filho, L. Ranula in children: Retrospective study of 25 years and literature review of the plunging variable. International journal of pediatric otorhinolaryngology 2021, 148, 110810, doi:10.1016/j.ijporl.2021.110810.).

Reviewer 2 Report

A truly intriguing approach to a common problem. The use of Hybnex could benefit other surgeons in similar situations. One comment: the authors use the terms acetaminophen and paracetamol interchangeably; please use only one to avoid confusion.

Author Response

Comments to the Authors:

- “A truly intriguing approach to a common problem. The use of Hybnex could benefit other surgeons in similar situations. One comment: the authors use the terms acetaminophen and paracetamol interchangeably; please use only one to avoid confusion.”

Response to the Reviewer:

We thank the reviewer to find our paper interesting. The typo has been fixed in the revised manuscript.

Reviewer 3 Report

Dear Authors,

the case is interesting but it should be clearly stated about Bioethic Committee agreement for this method of treatment if it is not as a standard procedure. Another point is to add more information about studies conducted on animals with Hybenx® gel to provide some evidences of the use. The term "marsupialization" should be explain in detailes. And the conclusions have to be revise because one case cannot be a base to tell about reproducibility of the use of Hybenx® gel. Please correct the manuscript according to this points.

Author Response

Comments to the Authors:

- “Dear Authors, the case is interesting, but it should be clearly stated about Bioethic Committee agreement for this method of treatment if it is not as a standard procedure.”

Response to the Reviewer:

We thank the reviewer to find our paper interesting. Hybenx is indicated for the decontamination of dental surfaces and the gingival walls of periodontal pockets. The use that has been made of it is conceptually the same but on a different pathological condition. That is, introduction of the Hybenx gel into an anatomical cavity bounded by mucous walls that has an external exit hole (as for a deep periodontal pocket). The use of Hybenx has been documented for the treatment of aphthous lesions (direct application on the injured mucosa) and periodontal and peri-implant pockets or periodontal acute abscess (introduction into the periodontal pockets). For this reason, being a contact action (the gel was removed after a few seconds, according to the manufacturer's instructions), it is not an off-label use, but a pathological condition never considered before. Furthermore, previous case reports using Hybenx gel did not refer to the ethics committee.

Comments to the Authors:

- “Another point is to add more information about studies conducted on animals with Hybenx® gel to provide some evidence of the use.”

Response to the Reviewer:

We thank the reviewer to take this aspect up. We added an animal study on the use of Hybenx gel in the revised manuscript (REF #20).

Comments to the Authors:

- “The term "marsupialization" should be explained in details.”

Response to the Reviewer:

Point taken. We have added the definition of “marsupialization” and “sclerotherapy” in the revised manuscript.

Comments to the Authors:

- “And the conclusions have to be revise because one case cannot be a base to tell about reproducibility of the use of Hybenx® gel. Please correct the manuscript according to this points.”

Response to the Reviewer:

The conclusion has been updated according to the reviewer’s instruction.

Round 2

Reviewer 1 Report

Dear authors,

Thank you for the review done.

Therefore, I found inconsistence in the explanation sent (responses).

“The use of Hybenx® gel for the treatment of pseudocystic lesions is a novelty that does not refer to existing publications that use the same method for the same purpose”

“The safety of Hybenx® gel, respecting the times and methods of application, is certified by the manufacturer. In this case, the same method recommended by the manufacturer for the decontamination of the subgingival root surfaces was used.”

How is there safety in the application (certificated by the manufacturer) if this use is a novelty?

Please, I want to request the IRB for this clinical case. Please, attach it in the supplementary material.

Author Response

Comments to the Authors:

-" Dear authors,

Thank you for the review done.

Therefore, I found inconsistence in the explanation sent (responses).

“The use of Hybenx® gel for the treatment of pseudocystic lesions is a novelty that does not refer to existing publications that use the same method for the same purpose”

“The safety of Hybenx® gel, respecting the times and methods of application, is certified by the manufacturer. In this case, the same method recommended by the manufacturer for the decontamination of the subgingival root surfaces was used.”

How is there safety in the application (certificated by the manufacturer) if this use is a novelty?

Please, I want to request the IRB for this clinical case. Please, attach it in the supplementary material.”

Response to the Reviewer:

We thank the reviewer for appreciating our effort for improving the paper. As suggested, we asked the IRB approval for this study. We added the IRB approval as supplementary material.

Reviewer 3 Report

There is an improvement of the work.

Please add to the text following information:

1. The manufacturer of Picibanil.

2. I did't find also information in the text about the agreement for the procedure by parents/caregiveres? Did they sign an information content?

3. It should be cleraly stated in the manuscript that this method is in accordance with ethical principles and for the benefit of the patient.

4. The surgery was done in a private dental office. Is this practice mentioned in the affiliation section?

Author Response

Comments to the Authors:

- “There is an improvement of the work.”

Response to the Reviewer:

We thank the reviewer for appreciating our effort in improving the manuscript.

Comments to the Authors:

- “Please add to the text following information:

  1. The manufacturer of Picibanil.”

Response to the Reviewer:

We thank the reviewer to take this aspect up. We added the manufacturer of Picibanil in the revised manuscript

Comments to the Authors:

- “2. I did't find also information in the text about the agreement for the procedure by parents/caregiveres? Did they sign an information content?

Response to the Reviewer:

We thank the reviewer to take this aspect up. We added the signature of the informed consent in the revised manuscript.

Comments to the Authors:

- “3. It should be cleraly stated in the manuscript that this method is in accordance with ethical principles and for the benefit of the patient.”

Response to the Reviewer:

Point taken. We have added the Institutional Review Board approval in the revised manuscript; as well, we also specified that the study was conducted in compliance with the ethical principles  of the Declaration of Helsinki.

Comments to the Authors:

- “4. The surgery was done in a private dental office. Is this practice mentioned in the affiliation section?”

Response to the Reviewer:

We thank the reviewer for the kind suggestion. The patient was sent from a private practice to the teaching hospital. We added the private practice in the affiliations of the revised manuscript.